

# Identifying vulgarity in Bengali social media textual content

Salim Sazzed

Computer Science, Old Dominion University, Norfolk, VA, USA

## ABSTRACT

The presence of abusive and vulgar language in social media has become an issue of increasing concern in recent years. However, research pertaining to the prevalence and identification of vulgar language has remained largely unexplored in low-resource languages such as Bengali. In this paper, we provide the first comprehensive analysis on the presence of vulgarity in Bengali social media content. We develop two benchmark corpora consisting of 7,245 reviews collected from YouTube and manually annotate them into vulgar and non-vulgar categories. The manual annotation reveals the ubiquity of vulgar and swear words in Bengali social media content (*i.e.*, in two corpora), ranging from 20% to 34%. To automatically identify vulgarity, we employ various approaches, such as classical machine learning (CML) classifiers, Stochastic Gradient Descent (SGD) optimizer, a deep learning (DL) based architecture, and lexicon-based methods. Although small in size, we find that the swear/vulgar lexicon is effective at identifying the vulgar language due to the high presence of some swear terms in Bengali social media. We observe that the performances of machine leanings (ML) classifiers are affected by the class distribution of the dataset. The DL-based BiLSTM (Bidirectional Long Short Term Memory) model yields the highest recall scores for identifying vulgarity in both datasets (*i.e.*, in both original and class-balanced settings). Besides, the analysis reveals that vulgarity is highly correlated with negative sentiment in social media comments.

## INTRODUCTION

Vulgarity or obscenity indicates the use of curse, swear or taboo words in language (*Wang, 2013*; *Cachola et al., 2018*). *Eder, Krieg-Holz & Hahn (2019)* conceived vulgar language as an overly lowered language with disgusting and obscene lexicalizations generally banned from any type of civilized discourse. Primarily, it involves the lexical fields of sexuality, such as sexual organs and activities, body orifices, or other specific body parts. *Cachola et al. (2018)* defined vulgarity as the use of swear/curse words. *Jay & Janschewitz (2008)* mentioned vulgar speech includes explicit and crude sexual references. Although the terms obscenity, swearing, and vulgarity have subtle differences in their meaning and scope, they are closely linked with some overlapping definitions. Thus, in this paper, we use them interchangeably to refer to the text that falls into the above-mentioned

Corresponding author
Salim Sazzed, ssazz001@odu.edu

definition of *Cachola et al. (2018)*, *Eder, Krieg-Holz & Hahn (2019)*, *Jay & Janschewitz (2008)*.

With the rapid growth of user-generated content in social media, vulgar words can be found in online posts, messages, and comments across languages. The occurrences of swearing or vulgar words are often linked with abusive or hatred context, sexism, and racism (*Cachola et al., 2018*), thus leading to abusive and offensive actions. Hence, identifying vulgar or obscene words has practical connections to understanding and monitoring online content. Furthermore, vulgar word identification can help to improve sentiment classification, as shown by various studies (*Cachola et al., 2018*; *Volkova, Wilson & Yarowsky, 2013*).

Social media platforms such as Twitter, Facebook, Instagram, YouTube have made virtual social interaction popular by connecting billions of users. In social media, swearing is ubiquitous according to various studies. *Wang et al. (2014)* found that the rate of swear word usage in English Twitter is 1.15%, almost double compared to its use in daily conversation (0.5%–0.7%) as reported by *Jay & Janschewitz (2008)*, *Mehl et al. (2007)*. *Wang et al. (2014)* also reported that 7.73% of tweets in their random sampling collection contain swear words. Based on *Jay & Janschewitz (2008)*, offensive speech can be classified into three categories: *vulgar*, which includes explicit and crude sexual references, *pornographic*, and *hateful*, which refers to offensive remarks targeting people's race, religion, country, etc. The categorization suggests that there exists a link between offensiveness and vulgarity.

Unlike English, research related to vulgarity is still unexplored in Bengali. As the vulgar word usage is dependent on the socio-cultural context and demography (*Cachola et al., 2018*), it is necessary to explore their usage in languages other than English. For example, the usage of f*ck, a*s, etc. are common in many English speaking countries in an expression to emphasize feelings, to convey neutral/idiomatic or even positive sentiment as shown by *Cachola et al. (2018)*. However, the corresponding Bengali words are highly unlikely to be used in a similar context in Bengali, due to the difference in the socio-culture of the Bengali native speakers (*i.e.*, people living in Bangladesh or India).

There is a lack of annotated vulgar or obscene datasets in Bengali, which are crucial for developing effective machine learning models. Therefore, in this work, we focus on introducing resources for vulgarity analysis in Bengali. Besides, we investigate the presence of vulgarity, which is often associated with abusiveness and inappropriateness in social media. Furthermore, we focus on automatically distinguishing vulgar comments (*e.g.*, usage of filthy language or curses towards a person), which should be monitored and regulated in online communications, and non-vulgar non-abusive negative comments, which should be allowed as part of freedom of speech.

We construct two Bengali review corpora consisting of 7,245 comments and annotate them based on the presence of vulgarity. We find a high presence of vulgar words in Bengali social media content based on the manual annotations. We provide the comparative performance of both lexicon-based and machine learning (ML) based methods for automatically identifying the vulgarity in Bengali social media data. As a lexicon, we utilize a Bengali vulgar lexicon, BengSwearLex, which consists of 184 swear and

obscene terms. We leverage two classical machine learning (CML) classifiers, Support Vector Machine (SVM) and Logistic Regression (LR), and an optimizer, Stochastic Gradient Descendent (SGD), to automatically identify vulgar content. In addition, we employ a deep learning architecture, Bidirectional Long Short Term Memory (BiLSTM). We observe that BengSwearLex provides a high recall score in one corpus and very high precision scores in both corpora. BiLSTM shows higher recall scores than BengSwearLex in both corpora in class-balanced settings; however, they yield high false positives, thus achieve a much lower precision score. The performances of the CML classifiers vary by the class distribution of the dataset. We observe that when the dominant class (*i.e.*, non-vulgar class) is undersampled to make the dataset class-balanced, CML classifiers attain much better performance. Class-balancing using over-sampling techniques like SMOTE (*Chawla et al., 2002*) or weighting classes based on the sample distributions fail to yield any significant performance improvement of CML classifiers in both datasets.

## Motivation

As vulgarity is often related to abusive comments, it is required to identify the presence of vulgarity on social media content. In Bengali, until now, no work has addressed this issue. Although a few papers tried to determine the offensive or hate speech in Bengali utilizing labeled data, none concentrated on recognizing vulgarity or obscenity. Since social media platforms such as Facebook, Twitter, YouTube, Instagram are popular in Bangladesh (the country with the highest number of Bengali native speakers), it is necessary to distinguish vulgarity in the comments or reviews for various downstream tasks such as abusiveness or hate speech detection and understanding social behaviors. Besides, it is imperative to analyze how vulgarity is related to sentiment.

## Contributions

The main contributions of this paper can be summarized as follows-

- We manually annotate two Bengali corpora consisting of 7,245 reviews/comments into vulgar and non-vulgar categories and make them publicly available (https://github.com/sazzadcsedu/Bangla-vulgar-corpus).
- We provide a quantitative analysis on the presence of vulgarity in Bengali social media content based on the manual annotation.
- We present a comparative analysis of the performances of lexicon-based, CML-based, and DL-based approaches for automatically recognizing vulgarity in Bengali social media content.
- Finally, we investigate how vulgarity is related to user sentiment in Bengali social media content.

## RELATED WORK

Researchers studied the existence and socio-linguistic characteristics of swearing, cursing, incivility or cyber-bullying in social media (*Wang et al., 2014*; *Sadeque et al., 2019*;

*Kurrek, Saleem & Ruths, 2020*; *Gauthier et al., 2015*; *Agrawal & Awekar, 2018*). *Wang et al. (2014)* investigated the cursing activities on Twitter, a social media platform. They studied the ubiquity, utility, and contextual dependency of swearing on Twitter. *Gauthier et al. (2015)* analyzed several sociolinguistic aspects of swearing on Twitter text data. *Wang et al. (2014)* investigated the relationship between social factors such as gender with the profanity and discovered males employ profanity much more often than females. Other social factors such as age, religiosity, or social status were also found to be related to the rate of using vulgar words (*McEnery, 2004*). *McEnery (2004)* suggested that social rank, which is related to both education and income, is anti-correlated to the use of swear words. The level of education and income are inversely correlated with the usage of vulgarity on social media, with education being slightly more strongly associated with the lack of vulgarity than income (*Cachola et al., 2018*). Moreover, liberal users tend to exercise vulgarity more on social media (*Cachola et al., 2018*; *Sylwester & Purver, 2015*; *Preotiuc-Pietro et al., 2017*).

*Eder, Krieg-Holz & Hahn (2019)* described a workflow for acquisition and semantic scaling of a lexicon that contains lexical items in the German language, which are typically considered as vulgar or obscene. The developed lexicon starts with a small seed set of rough and vulgar lexical items, and then automatically expanded using distributional semantics.

*Jay & Janschewitz (2008)* noticed that the offensiveness of taboo words depends on their context, and found that usages of taboo words in conversational context is less offensive than the hostile context. *Pinker (2007)* classified the use of swear words into five categories. Since many studies related to the identification of swearing or offensive words have been conducted in English, several lexicons comprised of offensive words are available in the English language. *Razavi et al. (2010)* manually collected around 2,700 dictionary entries including phrases and multi-word expressions, which is one of the earliest work offensive lexicon creations. The recent work on lexicon focusing on hate speech was reported by (*Gitari et al., 2015*).

*Davidson et al. (2017)* studied how hate speech is different from other instances of offensive language. They used a crowd-sourced lexicon of hate language to collect tweets containing hate speech keywords. Using crowd-sourcing, they labeled tweets into three categories: those containing hate speech, only offensive language, and those with neither. We train a multi-class classifier to distinguish between these different categories. They analyzed when hate speech can be reliably separate from other offensive language and when this differentiation is very challenging.

In Bengali, several works investigated the presence of abusive language in social media data by leveraging supervised ML classifiers and labeled data (*Ishmam & Sharmin, 2019*; *Banik & Rahman, 2019*). *Sazzed (2021)* annotated 3,000 transliterated Bengali comments into two classes, abusive and non-abusive, 1,500 comments for each. For baseline evaluations, the author employed several traditional machine learning (ML) and deep learning-based classifiers.

*Emon et al. (2019)* utilized linear support vector classifier (LinearSVC), logistic regression (LR), multinomial naïve Bayes (MNB), random forest (RF), artificial neural

network (ANN), recurrent neural network (RNN) with long short term memory (LSTM) to detect multi-type abusive Bengali text. They found RNN outperformed other classifiers by obtaining the highest accuracy of 82.20%. *Chakraborty & Seddiqui (2019)* employed machine learning and natural language processing techniques to build an automatic system for detecting abusive comments in Bengali. As input, they used Unicode emoticons and Unicode Bengali characters. They applied MNB, SVM, and Convolutional Neural Network (CNN) with LSTM and found SVM performed best with 78% accuracy. *Karim et al. (2020)* proposed BengFastText, a word embedding model for Bengali, and incorporated it into a Multichannel Convolutional-LSTM (MConv-LSTM) network for predicting different types of hate speech. They compared BengFastText against the Word2Vec (*Mikolov et al., 2013*) and GloVe (*Pennington, Socher & Manning, 2014*) embedding by integrating them into several ML classifiers.

However, none of the existing works primarily focused on recognizing vulgarity or profanity in Bengali social media data. To the best of our knowledge, it is the first attempt to identify and provide a comprehensive analysis of the presence of vulgarity in the context of Bengali social media data.

## SOCIAL MEDIA CORPORA

We create two vulgar datasets consisting of 7,245 Bengali comments. Both datasets are constructed by collecting comments from YouTube (https://www.youtube.com/), a popular social media platform.

### Drama review dataset

The first corpus we utilize is a drama review corpus. This corpus was created and deposited by *Sazzed (2020a)* for sentiment analysis; It consists of 8,500 positive and 3,307 negative reviews. However, there is no distinction between different types of negative reviews. Therefore, we manually annotate these 3,307 negative reviews into two categories; one category contains reviews that convey vulgarity, while the other category consists of negative but non-vulgar reviews.

### Subject-person dataset

The second corpus is also developed from YouTube. However, unlike the drama review corpus that represents the viewer's feedback regarding dramas, this corpus consists of comments towards a few controversial female celebrities.

We employ a web scraping tool to download the comment data from YouTube, which comes in JSON format. Employing a parsing script, we retrieve the comments from the JSON data. Utilizing a language detection library (https://github.com/Mimino666/langdetect), we recognize the comments written in Bengali. We exclude reviews written in English and Romanized Bengali (*i.e.*, Bengali language in the Latin script).

## CORPORA ANNOTATION

It is a common practice to engage multiple annotators to annotate the same entity, as it helps to validate and improve annotation schemes and guidelines, identifying ambiguities or difficulties, or assessing the range of valid interpretations (*Artstein, 2017*).

**Table 1 Annotation statistics of two raters (A1, A2) in drama review corpus.**

|  | Vulgar | Non-vulgar |
|---|---|---|
| Vulgar | 592 | 160 |
| Non-vulgar | 53 | 2,502 |

The comparison of annotations can be performed using a qualitative examination of the annotations, calculating agreement measures, or statistical modeling of annotator differences.

## Annotation guidelines

To annotate a corpus for an NLP task (*e.g.*, hate speech detection, sentiment classification, profanity detection), it is required to follow a set of guidelines (*Khan, Shahzad & Malik, 2021*; *Mehmood et al., 2019*; *Pradhan et al., 2020*; *Fortuna & Nunes, 2018*; *Sazzed, 2020a*).

Here, to distinguish the comments into vulgar and non-vulgar classes, annotators are asked to consider the followings guidelines-

- Vulgar comment: The presence of swearing, obscene language, vulgar slang, slurs, sexual and pornographic terms in a comment (*Eder, Krieg-Holz & Hahn, 2019*; *Cachola et al., 2018*; *Jay & Janschewitz, 2008*).
- Non-vulgar comment: The comments which do not have the above mentioned characteristics.

## Annotation procedure

The labeling is performed by three annotators (A1, A2, A3), who are Bengali native speakers; Among them, two are male and one female (A1: male, A2: female, A3: male). The first two annotators (A1 and A2) initially annotate all the reviews. Any disagreement in the annotation is resolved by the third annotator (A3).

## Annotation results

The annotations of a review by two reviewers (A1, A2) yield two scenarios.

1. Agreement: Both annotators (A1, A2) assign the same label to the review.
2. Conflict: Each annotator (A1, A2) assigns a different label to the review.

Table 1 presents the annotation statistics of the drama review dataset by two annotators (A1, A2). The Cohen's kappa ($\kappa$) (*Cohen, 1960*) provides an agreement score of 0.807, which indicates almost perfect agreement between two raters. Regarding the percentages, we find both reviewers agreed on 93.55% reviews.

In the subject-person dataset (Table 2), an agreement of 92.81% is observed between two annotators. Cohen's $\kappa$ yields a score of 0.8443, which refers to almost perfect agreement.

**Table 2  Annotation statistics of two raters (A1, A2) in subject-person dataset.**

|  | Vulgar | Non-vulgar |
|---|---|---|
| Vulgar | 1,282 | 120 |
| Non-vulgar | 163 | 2,373 |

**Table 3  Descriptions of two annotated corpora.**

| Dataset | Vulgar | Non-vulgar | Total |
|---|---|---|---|
| Drama | 664 | 2,643 | 3,307 |
| Subject-person | 1,331 | 2,607 | 3,938 |

| Drama Review Dataset | |
|---|---|
| বাংলা নাটকের গোয়া মোশাররফ করিম গং রাই মারতাছে, | Mosharraf Karims gang's are fucking Bengali drama, |
| চুদনাগিরি স্ক্রিপ্ট ছাড়া আর কোন স্ক্রিপ্ট ছিলো না।মাদারচোদ মার্কা নাটক এইটা | Wasn't there any other script except this fucking one. This is a motherfucker drama. |
| ব্লাইন্ডচোদ থানকির ছেলে। এতো অ্যাড চুদাও কে,,, | Fucker whore's son. why so many advertisements? |
| **Subject-Person Dataset** | |
| কুত্তার বাচ্চা তরে পাইলে দুইটা হাত কাটতাম নটি | Son of a Bitch, If I find you, I will chop your two hands, slut |
| শাহরিয়ার নাজিম ভাই থানকি নিয়া শো বন্ধ করুন | Sharir Nazim vai, please stop making tv show with whore |
| কিন্তু দুধের সাইজ বড়ো করে মনে হচ্ছে দুধ না ফুটবল | The enlarged tits look like a football, not tit |

**Figure 1  Sample vulgar reviews from annotated datasets.**

## Corpora statistics

After annotation the drama review corpus consists of 2,643 non-vulgar negative reviews and 664 vulgar reviews (Table 3). The presence of 664 vulgar reviews out of 3,307 negative reviews reveals a high presence (around 20%) of vulgarity in the dataset. The annotated subject-person dataset consists of 1,331 vulgar reviews and 2,607 non-vulgar reviews, a total of 3,938 reviews. This dataset contains even higher percentages of reviews labeled as vulgar, around 34%.

Figure 1 shows examples of vulgar reviews from drama and subject-person datasets. Figure 2 presents the top ten Bengali vulgar words and corresponding English translations for each dataset ('-' indicates corresponding English translation is not available due to language differences). We find a high presence of some vulgar words in the reviews, as shown in the top few rows. Besides, we observe a high number of misspelled vulgar words, which makes identifying them a challenging task. Among the top ten vulgar words in the subject-person dataset, we notice all of them except the last word (last row) are female-specific sexually vulgar terms. As the subjects of this dataset are female celebrities, this is expected. In the drama review dataset, among the top ten vulgar words, we find five terms as generic (not gender-specific) vulgar words, three are male-specific vulgar, and two are female-specific vulgar. The two female-specific vulgar terms also exist in the subject-person dataset.

| Bengali | English | Count |
|---|---|---|
| মাগি | Slut | 245 |
| দুধ | Tit | 181 |
| মাগির | Slut's | 180 |
| খানকি | Whore | 125 |
| মাগী | Slut | 93 |
| পতিতা | Prostitute | 69 |
| দুধের | Tit's | 47 |
| খানকি মাগি | Whore slut | 34 |
| খানকির | Whore's | 32 |
| বেশ্যা | Hooker | 30 |
| কুত্তা | Bitch | 29 |

| Bengali | English | Count |
|---|---|---|
| বালের | - | 235 |
| বাল | - | 66 |
| আবাল | Stupid | 27 |
| শালা | - | 26 |
| কুত্তার | Bitch | 23 |
| খানকির | Whore's | 21 |
| মাগির | Slut's | 14 |
| শালার | - | 14 |
| আচোদা | Fucking dumb | 13 |
| চোদা | Fuck | 12 |
| শাউয়ার | - | 11 |

**Figure 2 (A) Top 10 vulgar words in drama review dataset. (B) Top 10 vulgar words in subject-person dataset.** A dash indicates corresponding English translation is not available due to language differences.

## BASELINE METHODS

### Lexicon-based methods

We utilize two publicly available Bengali lexicons for identifying vulgar content in the text. The first lexicon we use is an obscene lexicon, *BengSwearLex* (https://github.com/sazzadcsedu/Bangla-Vulgar-Lexicon). The *BengSwearLex* consists of 184 Bengali swear and vulgar words, semi-automatically created from a social media *corpus*. The other lexicon is a sentiment lexicon, *BengSentiLex*, which contains around 1,200 positive and negative sentiment words (*Sazzed, 2020b*).

The purpose of utilizing a sentiment lexicon for vulgarity detection is to investigate whether negative opinion words present in the sentiment lexicon can detect vulgarity. The few other Bengali sentiment lexicons are dictionary-based word-level translations of popular English sentiment lexicons; thus, not capable of identifying swearing or vulgarity in Bengali text.

### Classical machine learning (CML) classifiers and SGD optimizer

Two popular CML classifiers, Logistics Regression (LR) and Support Vector Machine (SVM), and an optimizer, Stochastic Gradient Descendent (SGD), are employed to identify vulgar comments.

#### Input feature

We extract unigrams and bigrams from the text and calculate the term frequency-inverse document frequency (tf-idf) scores, which are used as inputs for the CML classifiers. The tf-idf is a numerical statistic that attempts to reflect the importance of a word in a document.

#### Parameter settings and library used

For LR (https://scikit-learn.org/stable/modules/generated/sklearn.linear_model.LogisticRegression.html) and SVM (https://scikit-learn.org/stable/modules/generated/sklearn.svm.SVC.html), the default parameter settings of scikit-learn library (*Pedregosa et al., 2011*) are used. For SGD, hinge loss and l2 penalty with a maximum iteration of

1,500 are employed. We use the scikit-learn library (*Pedregosa et al., 2011*) to implement the SVM, LR and SGD.

## Deep learning classifier

BiLSTM (Bidirectional Long Short Term Memory) is a deep learning-based sequence processing model that consists of two LSTMs (*Hochreiter & Schmidhuber, 1997*). BiLSTM processes input in both forward and backward directions, thus, provides more contextual information to the network.

### Network architecture, hyperparameter settings and library used

The BiLSTM model starts with the Keras embedding layer (*Chollet et al., 2015*). The three important parameters of the embedding layer are *input dimension*, which represents the size of the vocabulary, *output dimensions*, which is the length of the vector for each word, *input length*, the maximum length of a sequence. The *input dimension* is determined by the number of words present in a corpus, which vary in two corpora. We set the *output dimensions* to 64. The maximum length of a sequence is used as 200.

A drop-out rate of 0.5 is applied to the dropout layer; ReLU activation is used in the intermediate layers. In the final layer, softmax activation is applied. As an optimization function, Adam optimizer, and as a loss function, binary-cross entropy are utilized. We set the batch size to 64, use a learning rate of 0.001, and train the model for ten epochs. We use the Keras library (*Chollet et al., 2015*) with the TensorFlow backend for BiLSTM implementation.

# EXPERIMENTAL SETTINGS AND RESULTS

## Settings

### Lexicon-based method

If a review contains at least one term from BengSwearLex, it is considered as a vulgar. As BengSwearLex is comprised of only manually validated vulgar, slang, or swear terms, referring a non-vulgar comment to vulgar (*i.e.*, false positive) is highly unlikely; thus, a very high precision score close to one is expected.

### ML-based classifiers/optimizer

The results of ML classifiers are reported based on ten-fold cross-validation. We provide the performance of various ML classifiers in four different settings based on the class distribution,

1. Original setting: The original setting is class-imbalanced, as most of the comments in the dataset are non-vulgar.
2. Class-balancing using class weighting: This setting considers the distribution of the samples from different classes in the dataset. The weight of a class is set inversely proportional to its presence in the dataset.
3. Class-balancing using undersampling: In this class-balanced setting, we make use of all the samples from the minor class (*i.e.*, vulgar class). However, for the major class

(*i.e.*, non-vulgar class), from the entire sample, we randomly select a subset that has the same number of instances of the minor class.

4. Class-balancing using SMOTE: SMOTE is an oversampling technique that generates synthetic samples for the minor class (*Chawla et al., 2002*). It is used to obtain a synthetically class-balanced or nearly class-balanced set, which is then used to train the classifier.

## Evaluation metrics

We report the comparative performances of various approaches utilizing precision, recall and F1 measures. The $TP_V$, $FP_V$, $FN_V$ for the vulgar class $V$ is defined as follows-

$TP_V$ = true vulgar review classified as vulgar

$FP_V$ = true non-vulgar review classified as vulgar

$FN_V$ = true vulgar review classified as non-vulgar

The recall ($R_V$), precision ($P_V$) and F1 score ($F1_V$) of vulgar class are calculated as-

$$R_V = \frac{TP_V}{TP_V + FN_V} \tag{1}$$

$$P_V = \frac{TP_V}{TP_V + FP_V} \tag{2}$$

$$F1_V = \frac{2 * R_V * P_V}{R_V + P_V} \tag{3}$$

## Comparative results for identifying vulgarity

Table 4 shows that among the 664 vulgar reviews present in the drama review corpus, the sentiment lexicon identifies only 204 vulgar reviews (based on the negative score). The vulgar lexicon BengSwearLex registers 564 reviews as vulgar, with a high recall score of 0.85. In the original class-imbalanced dataset, all the CML classifiers achieve very low recall scores. However, when a class-balanced dataset is selected by performing undersampling to the dominant class, the recall scores of CML classifiers increase significantly to 0.90. However, we notice precision scores decrease in the class-balanced setting due to a higher number of false-positive (FP). BiLSTM provides the highest recall scores in both original and class-balanced setting, which is 0.70 and 0.94, respectively.

Table 5 shows the performances of various methods in subject-person dataset. We find that the sentiment lexicon shows a very low recall score, only 0.18. The BengSwearLex yields a recall score of 0.69. SVM, LR, and SGD exhibit low recall scores below 0.60 in the original class-imbalanced setting. However, in the class-balanced setting with undersampling (*i.e.*, 1,331 comments from both vulgar and non-vulgar categories), a higher recall score is observed. SGD yields a recall score of 0.77. BiLSTM shows the highest recall scores in both original and all the class-balanced settings, which is around 0.8. BiLSTM provides lower precision scores compared to CML classifiers in both settings (*i.e.*, original class-imbalanced and class-balanced).

**Table 4 Performance of various methods for vulgarity detection in drama review dataset.**

| Type | Method | # Identified vulgar review (out of 664) | $R_V$ | $P_V$ | $F1_V$ |
|---|---|---|---|---|---|
| Lexicon | *Sazzed (2020b)* | 204 | 0.307 | – | – |
|  | BengSwearLex | 564 | 0.849 | 0.998 | 0.917 |
| ML classifier (original setting) | LR | 161 | 0.245 | 1.0 | 0.394 |
|  | SVM | 345 | 0.534 | 0.994 | 0.686 |
|  | SGD | 386 | 0.588 | 0.985 | 0.736 |
|  | BiLSTM | 462 | 0.704 | 0.783 | 0.741 |
| ML classifier (undersampling) | LR | 609 | 0.917 | 0.801 | 0.855 |
|  | SVM | 593 | 0.893 | 0.859 | 0.876 |
|  | SGD | 592 | 0.891 | 0.876 | 0.883 |
|  | BiLSTM | 624 | 0.940 | 0.851 | 0.893 |
| ML classifier (SMOTE) | LR | 367 | 0.552 | 0.970 | 0.704 |
|  | SVM | 386 | 0.581 | 0.982 | 0.730 |
|  | SGD | 385 |  |  |  |
|  | BiLSTM | 563 | 0.850 | 0.707 | 0.772 |
| ML classifier (class weighting) | LR | 385 | 0.579 | 0.96 | 0.723 |
|  | SVM | 388 | 0.584 | 0.934 | 0.719 |
|  | SGD | 438 | 0.659 | 0.964 | 0.783 |
|  | BiLSTM | 564 | 0.854 | 0.667 | 0.749 |

**Table 5 Performance of various methods for vulgarity detection in subject-person dataset.**

| Type | Method | # Identified vulgar review (out of 1,331) | $R_V$ | $P_V$ | $F1_V$ |
|---|---|---|---|---|---|
| Lexicon | *Sazzed (2020b)* | 239 | 0.180 | – | – |
|  | BengSwearLex | 917 | 0.689 | 0.998 | 0.815 |
| ML classifiers (original setting) | LR | 551 | 0.394 | 0.992 | 0.563 |
|  | SVM | 788 | 0.594 | 0.962 | 0.746 |
|  | SGD | 860 | 0.660 | 0.940 | 0.775 |
|  | BiLSTM | 1050 | 0.793 | 0.724 | 0.757 |
| ML classifiers (undersampling) | LR | 954 | 0.717 | 0.870 | 0.786 |
|  | SVM | 969 | 0.728 | 0.893 | 0.802 |
|  | SGD | 1027 | 0.772 | 0.884 | 0.824 |
|  | BiLSTM | 1064 | 0.786 | 0.866 | 0.824 |
| ML classifier (SMOTE) | LR | 826 | 0.620 | 0.892 | 0.731 |
|  | SVM | 847 | 0.636 | 0.941 | 0.759 |
|  | SGD | 866 | 0.650 | 0.938 | 0.768 |
|  | BiLSTM | 1075 | 0.809 | 0.737 | 0.771 |
| ML classifier (class weighting) | LR | 911 | 0.684 | 0.814 | 0.743 |
|  | SVM | 824 | 0.619 | 0.912 | 0.737 |
|  | SGD | 935 | 0.702 | 0.904 | 0.790 |
|  | BiLSTM | 1070 | 0.807 | 0.742 | 0.773 |

| Bengali Comments | English Translation |
|---|---|
| বালের প্রেম ভালোবাসা সস্তা আবেগ ছাড়াও যে এত সুন্দর নাটক করা যায় তা আবারও দেখিয়ে দিলো বৃন্দাবন দা।অসাধারণ। | Vrindavan Da showed once again that it is possible to make such a beautiful drama apart from fucking love and cheap emotions. |
| জাস্ট অসাধারণ! ৯৩০ জন মাদারচোদ ডিসলাইক কি কারণে দিলো ওরা জানে! | Just awesome! I don't know what caused 930 motherfuckers to dislike! |
| কুত্তার বাচ্চা বলে যে সেটা মজার,,, আর শেষ খুব দারুণ | It says son of a bitch which is funny, and the end is great |

**Figure 3 Examples of positive reviews with vulgar words in drama review corpus.**

## Vulgarity and sentiment

We further analyze how vulgarity is related to user sentiment in social media. As a social media corpus, we leverage the entire drama review dataset, which contains 8,500 positive reviews in addition to 3,307 negative reviews stated earlier. Using the BengSwearLex vulgar lexicon, we identify the presence of vulgar words in the reviews. We perform a comparative analysis of the presence of vulgar words in both positive and negative reviews. We find only 37 positive reviews out of 8,500 positive reviews contain any vulgar words, which is only 0.4% of the total positive reviews. Out of 3,307 negative reviews, we observe the presence of vulgar words in 553 reviews, which is 16.67% of total negative reviews. Figure 3 shows examples of several positive reviews that contain vulgar terms.

## DISCUSSION

The results show that the sentiment lexicon yields poor performance in identifying vulgarity in Bengali textual content, as shown by its poor performance in both datasets. The poor coverage of the sentiment lexicon is expected as it contains different types of negative words, thus may lack words that are particularly associated with vulgarity. Besides, vulgarity is often linked with the usage of internet slang words that may not exist in small-sized sentiment lexicon.

The vulgar lexicon, BengSwearLex, on the other hand, provides a significantly higher recall scores than sentiment lexicon as it was specially curated to identify vulgarity, obscenity or swearing. The high presence of some of the vulgar words, as shown in Fig. 2 also helps BengSwearLex to achieve a good coverage (i.e., recall score) for vulgarity detection. We observe that the recall score of BengSwearLex varies in two corpora. In the smaller drama review data (664 vulgar review), it shows a recall score of 0.85, while in the other dataset which contains a much higher number of vulgar review, BengSwearLex achieve much lower recall score of 0.69. Since BengSwearLex contains less than 200 words, its performance can be affected by the characteristics and size of the dataset. BengSwearLex achieves almost a perfect precision score, close to one, in both corpora. Since BengSwearLex was manually validated to assure that it contains only vulgar or swear words, the almost perfect precision score is expected.

Tables 4 and 5 reveal that the performances of ML classifiers can be affected by the class distribution of the training data. Especially for the CML classifiers, when class-imbalanced training data is used, the results lean toward the dominating class (i.e., non-vulgar category) and achieve low recall and high precision scores, as shown in Tables 4 and 5.

Due to the presence of a much higher number of non-vulgar comments in the original dataset, CML classifiers yield a high number of false negatives (FN) and a low number of false positives (FP) for the vulgar class, which is reflected in the low recall score and high precision score. Whenever a class-balanced training set is employed, all the CML classifiers yield a higher recall score.

We find that the deep learning-based method BiLSTM is less affected by the class imbalance. Only when the difference of class proportion is very high, such as 18% *versus* 82% in the drama review dataset, we observe BiLSTM shows a high difference in recall score. Besides, we analyze the rationale behind using vulgar words in Bengali social media data. Although *Holgate et al. (2018)* found the usage of vulgar words can be non-offensive such as when used in informal communication between closely-related groups or expressing emotion in social media, we observe that when it is exercised in review or targeted towards a person with no personal connection, it is inappropriate or offensive most of the time.

## SUMMARY AND CONCLUSION

With the surge of user-generated content online, the detection of vulgar or abusive language has become a subject of utmost importance. While there have been few works in hate speech or abusive content analysis in Bengali, to the best of our knowledge, this is the first attempt to thoroughly analyze the existence of vulgarity in Bengali social media content.

This paper introduces two annotated datasets consisting of 7,245 vulgar reviews to address the resource scarcity in the Bengali language for vulgarity analysis. Besides, we investigate the prevalence of vulgarity in social media comments. Our analysis reveals a high presence of swearing or vulgar words in social media, ranging from 20% to 34% in two datasets. We explore the performance of various automatic approaches for vulgarity identification of Bengali and present a comparative analysis. The analysis reveals the strengths and weaknesses of varied approaches and provides the directions for future research.

### Funding
The authors received no funding for this work.

### Competing Interests
The authors declare that they have no competing interests.

### Author Contributions
- Salim Sazzed conceived and designed the experiments, performed the experiments, analyzed the data, performed the computation work, prepared figures and/or tables, authored or reviewed drafts of the paper, and approved the final draft.

## Data Availability

GitHub: https://github.com/sazzadcsedu/Bangla-vulgar-corpus.

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
