# Peer review of "Identifying vulgarity in Bengali social media textual content"

_PeerJ Computer Science, doi:10.7717/peerj-cs.665_

## Round 0.1 · original submission · Major Revisions

Reviewers have provided a set of detailed comments to flesh out the study (e.g. improve definition and operationalisation of vulgar content) and to make the experimentation more robust. I would advise the authors to address these suggestions through a round of major revisions.

Reviewer 1 ·

Basic reporting

The paper aim is to distinguish between vulgar/non-vulgar categories using lexicon-based and ML based approaches. The authors introduce a dat set of 7000 reviews and have annotators tag them for vulgarity at review level.

Deep learning is a type of machine learning, the paper presents them as different.

The first sentence in the abstract is wrong, at least according to the citations it mentions as a reference. In (Cachola et al 2018), vulgarity is defined at a word level, and is not used to express emotion in language. Actually, in (Holgate et al 2018), express emotion was just one out of 6 functions of vulgar words identified.

Another major issues is that slang is equated to obscene and vulgar words (27-28), which is not true e.g. 'luv' is a slang word, but is neither obscene or vulgar.

In general, it is unclear what the authors refer to by vulgarity at a sentence/review level throughout the paper and in the annotation task, as it seems the definition changes quite a bit (e.g. line 1 vs line 35, line 49).

Experimental design

SGD - that's not a ML method, is an optimization method for a tehnique like LR or SVM. I thin the author may be referring to SGD classifier function in sklearn, which optimizes a loss that can be associated to one of the above models.

Validity of the findings

The lexicon-based methods should be evaluated for precision as well, as these is a gold set for annotations, rather than assume precision is 1. If precision is indeed 1, then the F1 score for the bengvullex method is better than the ML methods on the drama review dataset and probably also best on the subject-person data set.

Based on these findings, the discussion section of the paper seems to be misguided.

The class imbalance can be handled through various approaches, including instance weighting and over/under-sampling.

For metrics, one could also use macro-F1, if we care about both classes equally.

Additional comments

Agreement between annotators is good, so dataset could be a useful resource if the authors make the concept they annotated clear (rather than vulgarity, it's probably more like offensiveness).

Based on this task definition, the work is not very novel. Previous work on related tasks was done even for Bengali in particular (https://arxiv.org/abs/2012.09686).

Due to its current popularity and good results, authors should have tried to train a BERT-based model.

·

Basic reporting

Generally the article is structured in a professional way. However, following are a few suggestions for improvement.

The citations are mixed up with running text causing the difficulty in reading. Please check the guidelines for authors to improve the citation format.

Use definite values (instead of writing "over" or "around" etc.): line 33, 52, 56, 74, 122, 156, 158, 171, 173, 219, 231, 233, 236, 253, 281. Why the accuracy is in a range (80%-90%)? It should be exact scores unless there's a technical reason for indefinite values?


There may be a separate section of "Annotation Guidelines".

Following are suggestions for improvement of language:
line 67: "As social ..." ==> "Since social ..."
line 76: "Analysis the presence" --- not understandable
line 98: "As many ..." --- missing "There are"
line 100: why "approximately" --- use exact figure if possible/known
line 105: "labeled data Ishmam..." --- missing any boundary/separator
line 188: "We" ==> "we"
Table 4: column 5 heading: "PVul" ==> "pVul"

Figure 1,2,3: add English transliteration and translation for understanding of reader

Formula for F1 score is not mentioned.
The columns of each table should be described in the text or in the caption.
The rows or cells having the highest scores should be BOLD in the result tables.

Experimental design

Methods described with sufficient detail except the following:

There is no information about separating Training and Test partitions.
If Training and Testing is same then it's certainly over-fitted result.
If it is due to cross-validation then this fact should be explicitly stated.

Validity of the findings

All underlying data have been provided and the conclusions are well stated.

Reviewer 3 ·

Basic reporting

Author addresses very interesting and challenging problem i.e., Identifying vulgarity in Bengali social media content. Overall paper is well written but here some comments that needs to be addressed.
Minor Comments
1. Add English description against each example of Bengali for better understanding.
2. Add 1 to 2 liner details of Machine Learning and Deep learning algorithms used.
3. Include more literature review on overall and resource poor languages like
a) Thomas Davidson, Dana Warmsley, Michael Macy, and Ingmar Weber. 2017. Automated hate speech detection and the problem of offensive language. In Eleventh international aaai conference on web and social media
b) Elisabeth Eder, Ulrike Krieg-Holz, and Udo Hahn. 2019. At the Lower End of Language—Exploring the Vulgar and Obscene Side of
German. In Proceedings of the Third Workshop on Abusive Language Online. 119–128
c) Paula Fortuna and Sérgio Nunes. 2018. A survey on automatic detection of hate speech in text. ACM Computing Surveys (CSUR) 51, 4
(2018), 1–30.
d) Khawar Mehmood, Daryl Essam, Kamran Shafi, and Muhammad Kamran Malik. 2019. Sentiment Analysis for a Resource Poor
Language—Roman Urdu. ACM Transactions on Asian and Low-Resource Language Information Processing (TALLIP) 19, 1 (2019), 1–15.

4. include appropriate references where appropriate like (Ron Artstein. 2017. Inter-annotator agreement. In Handbook of linguistic annotation. Springer, 297–313) for inter annotator agreement. also include references of ML algorithms

Major Comments:
Annotation guidelines are missing. Author may take inspiration from the following paper
a) Khan, Muhammad Moin, Khurram Shahzad, and Muhammad Kamran Malik. "Hate Speech Detection in Roman Urdu." ACM Transactions on Asian and Low-Resource Language Information Processing (TALLIP) 20, no. 1 (2021): 1-19.
b) Rahul Pradhan, Ankur Chaturvedi, Aprna Tripathi, and Dilip Kumar Sharma. 2020. A Review on Offensive Language Detection. In
Advances in Data and Information Sciences. Springer, 433–439.

Experimental design

Need to mention parameter and hyper parameters of algorithm used.
Details of word embedding is missing.
Also mention which libraries used in experiments like Tensorflow, keras etc

Validity of the findings

No Comments

---

## Round 0.2 · accepted · Accept

Based on my own read and that of one of the original reviewers, the scientific contribution of this paper is now deemed worthy of acceptance for publication.

As a final step, it is advised that figures 2 and 3 be fully translated into English to facilitate readability and ensure wider readership.

·

Basic reporting

The article has been improved and better written now.

The figure 1 has been translated completely, however the Figure 2 is translated partially and Figure 3 is not translated at all. It would more understandable for large body of readers if the translation of all non-English expressions is provided.

Experimental design

Improved.

Validity of the findings

Better.

Additional comments

Fine.